# Molecular Characterization, Expression Profiles of *SMAD4*, *SMAD5* and *SMAD7* Genes and Lack of Association with Litter Size in Tibetan Sheep

**DOI:** 10.3390/ani12172232

**Published:** 2022-08-30

**Authors:** Ruizhe Sun, Mingming Li, Na He, Xiaocheng Wen, Junxia Zhang

**Affiliations:** College of Agriculture and Animal Husbandry, Qinghai University, Xining 810016, China

**Keywords:** *SMAD4*, *SMAD5*, *SMAD7*, gene expression, litter size, Tibetan sheep

## Abstract

**Simple Summary:**

Selection of molecular markers used in the marker-assisted selection of the litter size of sheep is critically essential. In this study, we identified single nucleotide polymorphisms (SNPs), g.51537A>G in *SMAD5* and g.319C>T in *SMAD7*, from Tibetan sheep, and they were genotyped. The expression patterns of *SMAD4*, *SMAD5* and *SMAD7* in the tissues of Tibetan sheep were determined. Furthermore, we also performed an association analysis of the SNPs with litter size. The results indicated that *SMAD4*, *SMAD5* and *SMAD7* were extensively expressed in the various tissues of Tibetan sheep. Sequence analysis showed that *SMAD4* contained 1662 bp encoding 533 amino acids, *SMAD5* contained 1667 bp encoding 465 amino acids and *SMAD7* contained 1408 bp encoding 427 amino acids. Alignment analysis showed that the amino acid sequence of the three proteins shared high similarity with sheep, yak, cattle, dog, human, pig, chimpanzee, rhesus monkey and house mouse. Furthermore, association analysis showed that both g.51537A>G in *SMAD5* and g.319C>T in *SMAD7* were not significantly associated with litter size of Tibetan sheep.

**Abstract:**

SMAD4, SMAD5 and SMAD7 belonging to the transforming growth factor β (TGF-β) superfamily are indispensable for oocyte formation and development, ovarian organogenesis and folliculogenesis. However, only a few studies have investigated the characteristics of *SMAD4*, *SMAD5* and *SMAD7* in Tibetan sheep and the effect of their polymorphism on litter size. In this study, we examined the expression of *SMAD4*, *SMAD5* and *SMAD7* in 13 tissues of Tibetan sheep by reverse transcription-quantitative polymerase chain reaction. Further, cDNA of these genes was cloned, sequenced and subjected to bioinformatics analysis. DNA sequencing was also used to detect single nucleotide polymorphisms (SNPs). However, iM-LDR^TM^ technology was used for SNP genotyping. Associations between polymorphisms and litter size were analyzed using data from genotyping of 433 Tibetan sheep. The results showed that the expression of *SMAD4*, *SMAD5* and *SMAD7* genes was ubiquitous in the tissues of Tibetan sheep, such as the ovary, uterus and oviduct, hypothalamus, hypophysis, heart, liver, spleen, lung, kidney, rumen, duodenum and *longissimus dorsi*. However, the expression was unbalanced and upregulated in the spleen, lung, ovary and uterus and downregulated in the *longissimus dorsi*. The bioinformatics analysis showed that *SMAD4*, *SMAD5* and *SMAD7* in Tibetan sheep encoded proteins of 533, 465 and 427 amino acids, respectively. Sequence homology analysis of the three proteins among other animals showed that the sequences of *SMAD4*, *SMAD5* and *SMAD7* of Tibetan sheep were similar to those in sheep, yak, cattle, dog, human, pig, chimpanzee, rhesus monkey and house mouse. Two synonymous mutations, g.51537A>G and g.319C>T, were detected in *SMAD5* and *SMAD7*, respectively. The associations of these SNPs and litter size were determined, and it was found that both g.51537A>G and g.319C>T have no significant effect on the litter size of Tibetan sheep. The results provided novel insights into the molecular characterization, expression profiles and polymorphisms of *SMAD4*, *SMAD5* and *SMAD7* in Tibetan sheep, but our results do not support associations between these genes and the litter size of Tibetan sheep.

## 1. Introduction

The Qinghai–Tibet Plateau (Tibetan Plateau) in China is the largest and highest plateau in the world [1]. It is located in northwest China and is characterized by extreme environmental conditions and a unique ecological environment involving high altitude, low oxygen levels, strong ultraviolet rays, low temperature, less rainfall and complex and variable terrain [2,3,4,5] inhabited by the numerous domestic animals. These animals play an essential role in the ecosystem and the livelihoods of the plateau residents. Many domestic animals have lived for generations and have stable genetic adaptability to harsh living conditions [3]. Tibetan sheep (*Ovies aries*) is one of the domestic animal breeds distributed on the plateau. After long-term natural selection, Tibetan sheep are well-adapted to the high-altitude region. They are the most numerous livestock in the Qinghai–Tibet Plateau [1]; however, they have low reproductive and production rates. 

The transforming growth factor-β (TGF-β) superfamily of cytokines participates in cellular proliferation, survival and differentiation [5]. The members of TGF-β include TGF-β activins, bone morphogenetic proteins (BMPs), growth differentiation factors (GDFs), glial-derived neurotrophic factors (GDNFs), nodal, lefty and anti-Müllerian hormones (AMH) [6,7,8,9]. TGF-β signaling is mediated by TGF-β ligands that transduce the signals to the nucleus via the action of SMADs [10]. The different TGF-β superfamily members signal via different SMAD proteins. Thus far, eight mammalian SMAD proteins have been isolated and designated from SMAD1 to SMAD8 [11]. Based on their function, SMADs are classified into the following three groups that exert their effects: receptor-activated SMADs (R-SMADs: SMAD1, SMAD2, SMAD3, SMAD5 and SMAD8), common mediator SMADs (Co-SMADs: SMAD4) and inhibitory SMADs (I-SMADs: SMAD6 and SMAD7) [10]. The SMAD proteins transduce signals in a dynamic process: they continually shuttle between the cytoplasm and nucleus, and ligand stimulation tunes this process. SMAD4 is the only co-mediated SMAD (Co-SMAD) in mammals. SMAD4 cannot be phosphorylated by the receptors; therefore, they are oligomerized with activated R-SMADs and form complexes via the regulation of p21 and other downstream genes for cell cycle arrest in the G1 phase and inhibition of cell proliferation [12]. SMAD5, which belongs to R-SMAD, is activated by BMP ligands and is phosphorylated by type I receptors [13], which are highly homologous in structure, but their functions are not the same [12]. SMAD7 is induced by TGF-β family members and acts as negative feedback by interfering with R-SMADs for receptor interaction and marking the receptors for degradation [6,14]. It generally antagonizes growth inhibition of the cell, matrix formation, apoptosis induction and embryonic lung formation [12]. The individual SMAD proteins influence proper functioning of the ovary [15]. As reported, *SMADs* are ubiquitously expressed in nearly all cell types in the body [6]. In the ovary, several TGF-β superfamily members are expressed by the oocyte, granulosa and thecal cells at different stages of folliculogenesis, although it has been demonstrated that *SMADs* are related to the activity of the ovary [16]. Currently, various candidate genes in the TGF-β signaling pathway have been identified, and several genes, which, to a greater or lesser extent, affect ewe prolificacy, have been identified in studies. *FecB* mutation (c.746 A>G) in *BMPR-IB* (bone morphogenetic protein receptor IB) [17] and the mutations in *BMP15* (bone morphogenetic protein 15) [18] and *GDF9* (growth differentiation factor 9) [19] have been identified as the major genes controlling prolificacy in sheep; however, the association between *SMAD4*, *SMAD5* and *SMAD7* genes and litter size of Tibetan sheep remains unclear. Therefore, this study aimed to clone *SMAD4*, *SMAD5* and *SMAD7* genes, detect polymorphism and perform association analysis for litter size in Tibetan sheep. Moreover, expression profiles of these genes were measured in different tissues.

## 2. Materials and Methods

### 2.1. Animals and Sampling

Tibetan sheep were obtained from Xiangkemeiduo Sheep Industry Co. Ltd. in Qinghai, China, and a total of 433 ewes were selected for the experiment. During the raising period, they were pastured in similar grassland conditions, and environmental conditions were kept consistent for all Tibetan sheep. The health and reproduction records of the ewes were kept by the farmers. Their mean litter size was calculated from 3 consecutive records from the first to the third parity (Appendix A). Finally, under the supervision of qualified veterinarians, blood samples (5 mL) were taken from the jugular vein of each sheep in the morning for DNA extraction. DNA was extracted using an EasyPure Blood Genomic DNA Kit (TransGen Biotech, Beijing, China). DNA was dissolved in Elution Buffer (10 mM Tris-HCl, 1 mM EDTA; pH 8.0) and stored at −20 °C.

A total of three ewes of Tibetan sheep were selected from the purebred herds of the same farm. The selected 6-month-old ewes were healthy and had similar weight. The ewes were slaughtered, and tissues were obtained from the hypothalamus, hypophysis, heart, liver, spleen, lung, kidney, ovary, oviduct, uterus, rumen, duodenum and *longissimus dorsi*. The tissue samples were immediately frozen in liquid nitrogen and then stored at −80 °C in a cryogenic freezer. The RNA from tissues was extracted using TransZol (TransGen Biotech, Beijing, China). RNA integrity, concentration and purity were measured by electrophoresis in 1% agarose gel and Nanodrop spectrophotometry(Gel DocTM XR+, Thermo Fisher Scientific Inc., Waltham, MA, USA), respectively. Only complete and clear bands of RNAs representing a 260 nm/280 nm ratio between 1.8 and 2.0 were used in this study. Then, the total RNA for each tissue was reverse-transcribed to cDNA by TransScript One-Step gDNA Removal (TaKaRa, Dalian, China). RNA integrity, concentration and purity were measured by electrophoresis in 1% agarose gels and Nanodrop spectrophotometer (Gel Doc TMXR+, Thermo Fisher Scientific Inc.), respectively. Only complete and clear bands of RNAs representing a 260nm/280 nm ratio between 1.8 and 2.0 were used in this study.

### 2.2. cDNA Cloning and Sequence Analysis

PCR amplification was performed using the synthesized cDNA as a template with several pairs of degenerate primers, which were designed using the coding regions of the 3 genes (Table 1). PCR program was as follows: 94 °C for 5 min, 30 cycles of 94 °C for 30 s, T_m_ °C for 30 s and 72 °C for 30 s, followed by one cycle at 72 °C for 5 min. PCR products were electrophoresed in a 1% (*w*/*v*) agarose gel and visualized by staining with Goldview Nucleic Acid Dye (Goldview type I). The DNA fragments were purified using an Agarose Gel DNA Purification Kit (TaKaRa, Dalian, China), further cloned into a pMD19-T vector (volume of 10 µL of 50 ng DNA, 50 ng pMD19-T vector, 5 µL Solution I, incubated at 4 °C overnight). They were finally transformed into *Escherichia coli* DH5α (Takara) competent cells and grown in Luria–Bertani agar plates with Ampicillin. White colonies were selected. The positive clones were sequenced by Shanghai Sangon Biological Engineering Company (China). Alignments of multiple sequences were obtained by BLAST analyses. DNAMAN software (version 6.0, Lynnon Biosoft, Montreal, QC, Canada) was used to determine the homologies of SMAD4, SMAD5 and SMAD7 proteins from different species, and the phylogenetic trees were constructed using the N-J method of MEGA7. The secondary structure prediction of Tibetan sheep SMAD4, SMAD5 and SMAD7 proteins was performed by the online protein analysis system SOPMA. The specific 3D structures (tertiary structures) were predicted using SWISS-MODEL.

### 2.3. Gene Expression Analysis 

*SMAD4*, *SMAD5* and *SMAD7* primers for real-time quantitative-polymerase chain reaction (qPCR) were designed according to NM_001267886.1, XM_004008815.3 and XM_027961075.1, respectively (Table 1). Real-time PCR was performed at 95 °C for 15 min, followed by 95 °C for 10 s, 50 °C for 32 s for 40 cycles and 72 °C for 30 s. qPCR was performed using a CFX96 Touch Real-Time PCR (BIO-RAD, Hercules, CA, USA). *GAPDH* was used as an internal reference to normalize target gene expression. All experiments were performed in triplicate. 

### 2.4. SNP Identification and Genotyping

Using the dbSNP database (http://www.ncbi.nlm.nih.gov/snp, accessed on 4 May 2022), *SMAD4*, *SMAD5* and *SMAD7* genes’ SNPs were screened and verified by DNA sequencing. Improved multiplex ligation detection reaction (iMLDR^TM^) was used for genotyping following the instrument operating guidelines. 

### 2.5. Statistical Analysis

The relative gene expression was calculated according to the 2^−^^ΔΔ^*^Ct^* method [20], where Δ*Ct = Ct*
_(__target gene__)_ − *Ct _(_**_GAPDH_*_)_, and 2^−^^ΔΔ^*^Ct^* indicates fold change in gene expression of the *SMAD4*, *SMAD5* and *SMAD7* genes’ mRNA relative to *GAPDH* gene. The data were analyzed with statistical software SPSS 19.0. One-way ANOVA was used to calculate differences in expression among different tissues.

The association analysis between genotypes and litter size of ewes was determined according to a general linear model (GLM) program. Based on the characteristics of sheep, the statistical model was as follows:*Y_ijn_* = *μ* + *P_i_* + *G_j_* + *I_PG_* + *e_ijn_*

where *Y_ijn_* is the phenotypic observation value; *μ* is the overall population mean; *P_i_* is the fixed effect of the *i*th parity (*i* = 1, 2 or 3); *G_j_* is the effect of the *j*th genotype (*j* = 1, 2 or 3); *I_PG_* is the interactive effect of parity and genotype and *e_ijn_* represents random error.

## 3. Results

### 3.1. Expression Profiles of Genes in Sheep

The RT-qPCR was used to investigate the general tissue distributions of the *SMAD4*, *SMAD5* and *SMAD7* genes in Tibetan sheep. The results showed that three genes were widely expressed (Figure 1). They were detected in all thirteen tissues, including heart, liver, spleen, lung, kidney, rumen, duodenum, hypothalamus, pituitary, *longissimus dorsi*, oviduct, ovary and uterus tissues. *SMAD4* gene expression was significantly higher in uterus than in other tissues (*p* < 0.05), and it was significantly higher in spleen and lung than in heart, liver, spleen, lung, kidney, rumen, duodenum, hypothalamus, pituitary, *longissimus dorsi*, oviduct and ovary tissues (*p* < 0.05). *SMAD5* gene expression was the highest in ovary (*p* < 0.05), followed by lung and spleen (*p* < 0.05). *SMAD7* gene expression was significantly higher in lung than in other tissues (*p* < 0.05).

### 3.2. cDNA Cloning and Bioinformatic Analysis of Genes

In this study, 1662 bp of Tibetan sheep *SMAD4* gene was cloned. Sequencing analysis showed that the ORF of *SMAD4* encodes a protein of 553 amino acids with a predicted molecular weight (MW) of 60.5 kD and isoelectric point (*p*I) of 7.63. A 1667 bp of *SMAD5* sequence was cloned. Subsequent sequencing analysis showed that the ORF of *SMAD5* contains 1398 bp and encodes a protein of 465 amino acids with a predicted molecular weight (MW) of 52.20 kD and isoelectric point (*p*I) of 7.63. After the Tibetan sheep, the *SMAD7* gene was cloned and a 1408 bp sequence of *SMAD7* was attained, containing 1284 bp ORF. The predicted Tibetan sheep SMAD7 protein consists of 427 amino acids with a molecular weight of 46.50 kDa and an isoelectric point of 8.63. The homology analysis of SMAD4 proteins from different species showed that Tibetan sheep have 100%, 100%, 99.8%, 99.6%, 99.3%, 99.3%, 99.3%, 99.3%, 90.0%, 74.8% and 63.6% sequence similarity with the SMAD4 sequences of sheep, yak, cattle, dog, human, pig, chimpanzee, rhesus monkey, house mouse, chicken and zebra, respectively (Figure 2A). The predicted Tibetan sheep SMAD5 protein was aligned with its homologues in other species. The results showed that the sequence of Tibetan sheep SMAD5 has 100.0%, 99.6%, 99.6%, 99.6%, 99.6%, 99.6%, 99.6%, 99.6%, 98.5%, 97.2% and 90.7% identity with its counterparts in sheep, yak, cattle, dog, human, pig, chimpanzee, rhesus monkey, house mouse, chicken and zebra, respectively (Figure 2B). The sequence of SMAD7 proteins in Tibetan sheep has 100.0%, 99.1%, 98.8%, 98.4%, 98.1%, 98.1%, 97.9%, 97.4%, 97.4%, 82.4% and 81.1 identity with its homologues in sheep, yak, pig, human, cattle, rhesus monkey, chimpanzee, house mouse, dog, chicken and zebra, respectively (Figure 2C). The phylogenetic trees were constructed (Figure 3A–C), and they showed that Tibetan sheep *SMAD4*, *SMAD5* and *SMAD7* are positioned in one clade with sheep *SMAD4*, *SMAD5* and *SMAD7*, respectively, indicating that these are the most closely related homologs, followed by yak, cattle and zebra, which has the farthest relationship with Tibetan sheep.

The secondary structure prediction of Tibetan sheep SMAD4, SMAD5 and SMAD7 proteins was performed (Figure 4). The results showed that the extension chain of sMAD4 composed of alpha-helix, extended strand, beta turn and random coil accounted for 53.35%, 24.41%, 15.55% and 6.69%, respectively, and the secondary structure components of the SMAD5 contained 57.63% alpha-helix, 21.51% extended strand, 16.99% beta turn and 3.87% random coil, and, for SMAD7 protein, alpha-helix, extended strand, beta turn and random coil were 33.76%, 15.92%, 3.82% and 46.50%, respectively. 

The specific 3D structures (tertiary structure) of Tibetan sheep SMAD4, SMAD5 and SMAD7 domain areas were predicted (Figure 5). The results showed that all the SMAD4, SMAD5 and SMAD7 domain areas constituted curved spiral structures. The 3D structures of SMAD4, SMAD5 and SMAD7 proteins are consistent with the prediction of secondary structures.

### 3.3. Polymorphism of Genes in Sheep 

Sequencing analysis identified two SNPs in *SMAD5* and *SMAD7* genes in Tibetan sheep. The g.51537A>G was detected in *SMAD5*, located in exon 3 and belonging to the synonymous type. We genotyped the identified SNPs of g.51537A>G in the *SMAD5* gene. Three genotypes were found, namely *AA*, *AG* and *GG*. The genotype frequency of *AA*, *AG* and *GG* was 0.59, 0.35 and 0.06, respectively. The allele frequencies of A and G were 0.76 and 0.24. The observed homozygosity (*Ho*), heterozygosity (*He*) and effective allele numbers (*Ne*) were 0.64, 0.36 and 1.56, respectively. The polymorphism information content (*PIC*) analysis showed that the SNPs were in moderate diversity (0.25 *< PIC <* 0.5). The χ^2^ test showed that identified SNP was in Hardy–Weinberg equilibrium. We screened out the g.319C>T in the *SMAD7* gene of Tibetan sheep, which was located in exon 3 and belonged to synonymous types. Only two genotypes were identified: *CC* and *CT,* with frequencies of 0.94 and 0.06, respectively. The allele frequencies of C and T were 0.97 and 0.03. *Ho*, *He* and *Ne* were 0.95, 0.05 and 1.06, respectively. The *PIC* analysis indicated the SNPs in low diversity (*PIC <* 0.25). The identified SNP was in Hardy–Weinberg equilibrium. The distribution, genotype and allele frequencies of the two identified SNPs were presented in Table 2, and the population genetic analysis of the two SNPs was listed in Table 3.

### 3.4. Association between the Snp Loci of Genes and Litter Size of Tibetan Sheep

Possible associations between the SNPs and litter size were investigated in Tibetan sheep populations (Table 4). This analysis revealed that the different genotypes of *SMAD5* and *SMAD7* and litter size of Tibetan sheep had a lack of association. No significant difference was found between the litter size of individuals carrying different genotypes in the g.51537A>G site of *SMAD5* (*p* > 0.05). Yet, no significant difference was found between the litter size of individuals carrying the *CC* and *CT* genotypes in g.319C>T of *SMAD7* (*p >* 0.05).

## 4. Discussion

The SMAD family of proteins is the key component of TGF-β family signal transduction in cells [21]. In the present study, we mainly studied three genes: *SMAD4*, *SMAD5* and *SMAD7*, and focused on the expression profiles, molecular characterization, polymorphism and association with the litter size of these genes in Tibetan sheep. The tissue expression profile revealed that *SMAD4*, *SMAD5* and *SMAD7* in Tibetan sheep exhibited broad expression patterns, including their expression in the hypothalamus, hypophysis, heart, liver, spleen, lung, kidney, ovary, oviduct, uterus, rumen, duodenum and *longissimus dorsi*. The *SMAD4*, *SMAD5* and *SMAD7* mRNA levels were unevenly distributed within all thirteen tissues. The *SMAD4*, *SMAD5* and *SMAD7* expressions were higher in the lung, spleen and uterus or ovary tissues than in other tissues, and their lowest expressions were in the *longissimus dorsi* tissues. The members of the SMAD pathway are closely associated with reproduction and follicular development, regulating ovulation and the litter size of sheep [15]. Xu reported that *SMAD4* and *SMAD7* mRNAs were present in the ovary and other tissues, including the hypothalamus, pituitary, uterus, heart, liver, spleen, lung, kidney, muscle and oviduct of Hu sheep [22]. Rodriguez et al. uncovered that SMAD1/5/4-mediated signaling is indispensable for the structural and functional integrity of the oviduct and uterus. The loss of SMAD1/5/4 alters the development of the oviduct smooth muscle layer, impeding embryo transit [19]. *SMAD4* expression was ubiquitous within the hypothalamus, pituitary, ovary, oviduct and uterus of small tail Han sheep and yaks [23,24]. *SMAD5* was expressed in the heart, liver, spleen, lung, kidney, rumen, reticulum, omasum, abomasum, small intestine, large intestine, subcutaneous fat and *longissimus dorsi* tissues of Qinchuan cattle [6]. *SMAD7* was expressed in domestic pigs’ brains, cerebellum, hypothalamus, kidney, liver and lung tissues; however, its expression was low in the heart and small intestine tissues and was almost not expressed in muscle tissues [25]. Our experimental results were generally consistent with the above-reported results. *SMAD4*, *SMAD5* and *SMAD7* are ubiquitously expressed in nearly all cell types in the body. Therefore, the three genes may play a prominent role as ovary-specific SMAD-interacting proteins in TGF-β superfamily ligand-target-gene selection in the ovary [26]. 

In this study, the sequences of *SMAD4*, *SMAD5* and *SMAD7* in Tibetan sheep were cloned. The multiple sequence alignment analysis indicated that the protein sequences of SMAD4, SMAD5 and SMAD7 in the sheep were similar to those in other mammals (greater than 90% sequence homology for most analyzed species), with the greatest similarity with sheep for SMAD4, SMAD5 and SMAD7, which is consistent with the results of the phylogenetic analysis. These results showed that SMAD4, SMAD5 and SMAD7 are highly conserved among different species.

Understanding protein structures is essential for determining their functions [27]. DNA sequences are translated into amino acid sequences, and researchers determine their functions either by comparing protein sequences with each other or by comparing their specific 3D structures [28]. In the present study, the results showed all the SMAD4, SMAD5 and SMAD7 domain area structures. A clear understanding of their structures will facilitate further research on understanding SMAD functions in the ovary. 

In this study, two SNPs from the candidate genes *SMAD5* and *SMAD7* were screened out to analyze the effects of SNPs on the litter size of Tibetan sheep. The g.51537A>G in *SMAD5* and the g.319C>T in *SMAD7* were synonymous changes, which could affect gene function by mRNA processing and protein translation and folding [29,30]. Thus, we hypothesized that the synonymous mutations might lead to the change in phenotype and affect litter size of sheep; nevertheless, association analysis revealed that the two SNPs of two genes were not associated with litter size in Tibetan sheep. Whether the two SNPs have no effect on the phenotype trait of litter size in other breeds of sheep remains to be further investigated.

## 5. Conclusions

In the present study, we showed that *SMAD*4, *SMAD5* and *SMAD7* genes were not only expressed in reproduction organs (ovary, uterus and oviduct) but also in other tissues: hypothalamus, hypophysis, heart, liver, spleen, lung, kidney, rumen, duodenum and *longissimus dorsi*, with the lowest expression in *longissimus dorsi*. *SMAD4*, *SMAD5* and *SMAD7* genes of Tibetan sheep were cloned and subjected to bioinformatics analysis. Based on the results of these experiments, *SMAD4*, *SMAD5* and *SMAD7* encoded 533, 465 and 427 amino acids, respectively. Additionally, the homology of three proteins between Tibetan sheep and other species is high. Two synonymous SNPs in the *SMAD5* and *SMAD7* genes were detected using DNA sequencing and then genotyped using iMLDR^TM^ methods. The association analysis revealed that the two SNPs were not significantly associated with litter size in the Tibetan sheep in this study.

## Figures and Tables

**Figure 1 animals-12-02232-f001:**
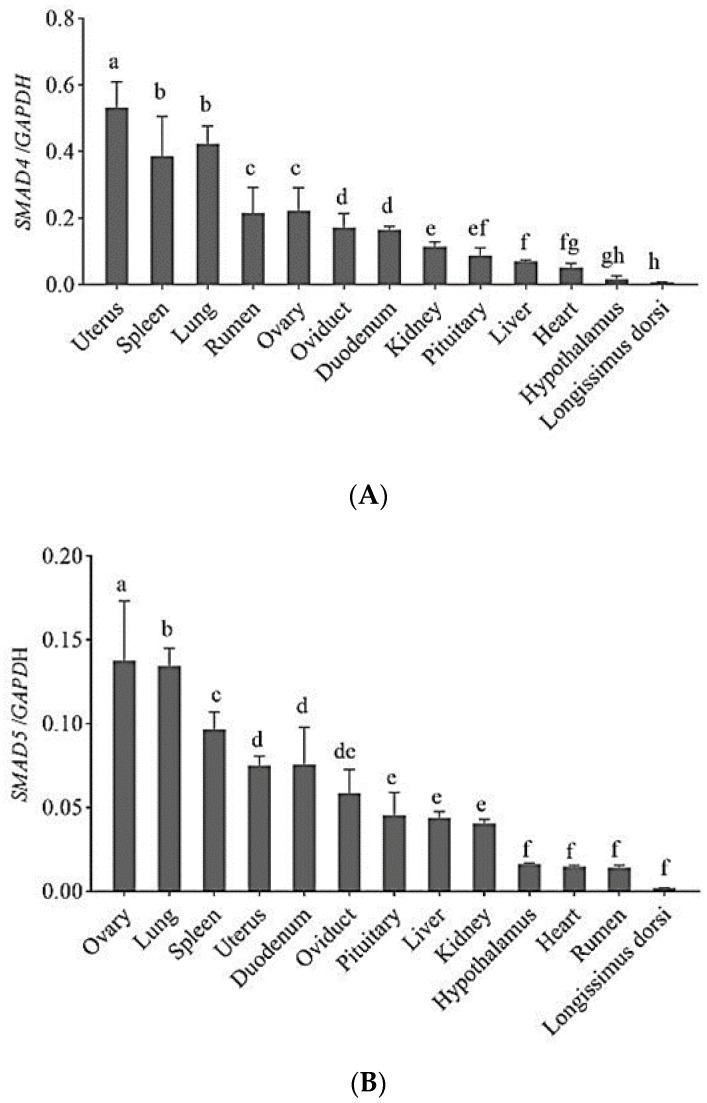
Expression of *SMAD4* (**A**), *SMAD5* (**B**) and *SMAD7* (**C**) mRNAs in different tissues of Tibetan sheep. The expression of *SMAD4*, *SMAD5* and *SMAD7* genes was detected by qPCR, and the *GAPDH* gene was used as the internal control gene. Each sample was repeated to determine in triplicate, and the data were collected as mean ± standard deviation. Note: different superscripts indicate significant difference (*p* < 0.05).

**Figure 2 animals-12-02232-f002:**
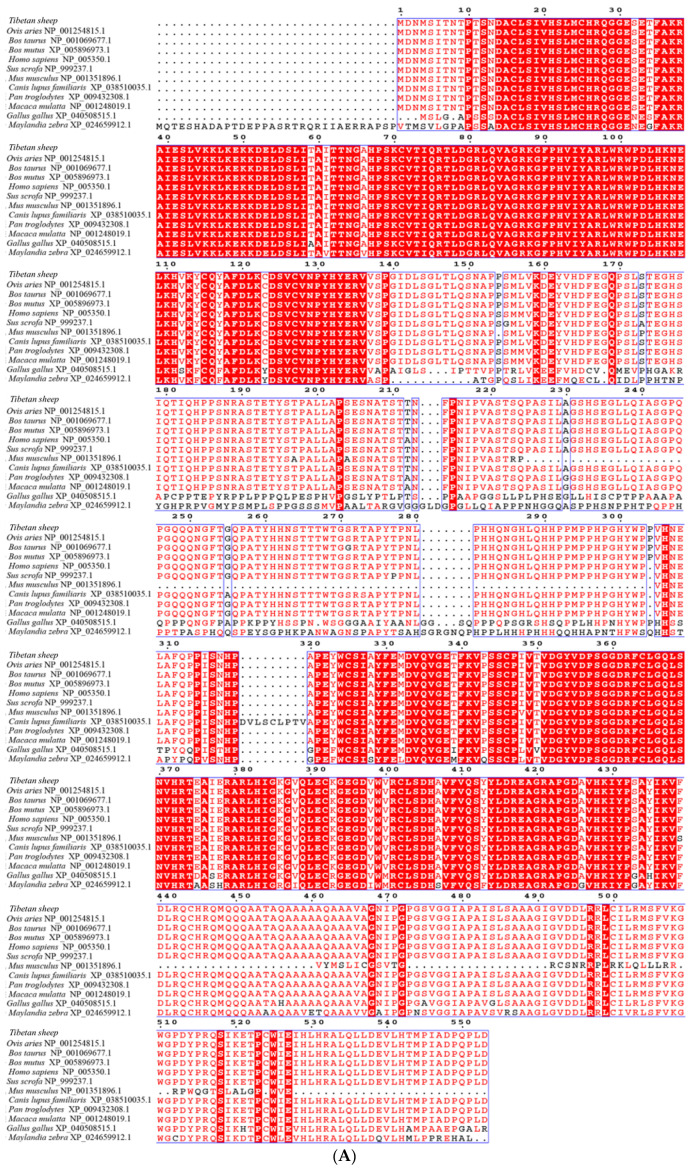
Amino acid sequence alignments of SMAD4 (**A**), SMAD5 (**B**) and SMAD7 (**C**) form *Ovis aries*. *Bos Taurus*, *Bos mutus*, *Homo sapiens*, *Sus scrofa*, *Mus musculs*, *Maylandia zebra*, *Pan troglodytes*, *Macaca mulatta*, *Gallus gallus* and *Canis lupus familiaris*.

**Figure 3 animals-12-02232-f003:**
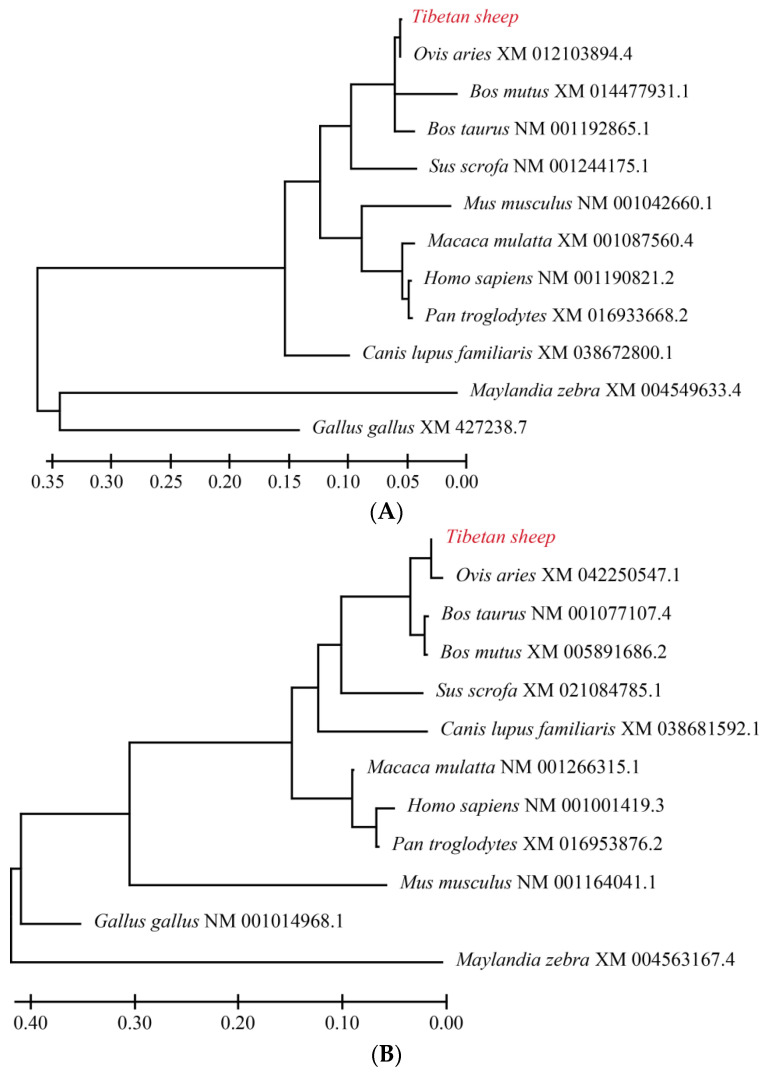
Phylogenetic tree of *SMAD4* (**A**), *SMAD5* (**B**) and *SMAD7* (**C**) of Tibetan sheep, sheep, cattle, yak, house mouse, human, chimpanzee, rhesus monkey, pig, dog, chicken and zebra. Red word represents the sequence of Tibetan sheep.

**Figure 4 animals-12-02232-f004:**
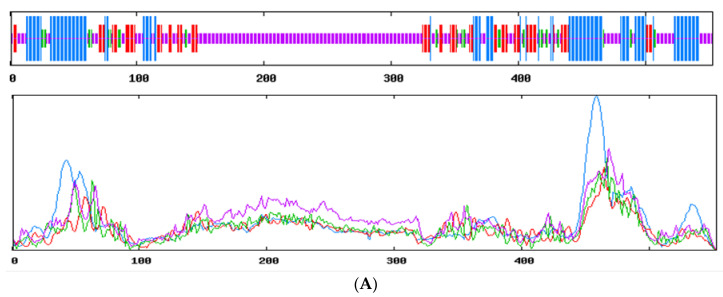
Secondary structure of SMAD4 (**A**), SMAD5 (**B**) and SMAD7 (**C**).

**Figure 5 animals-12-02232-f005:**
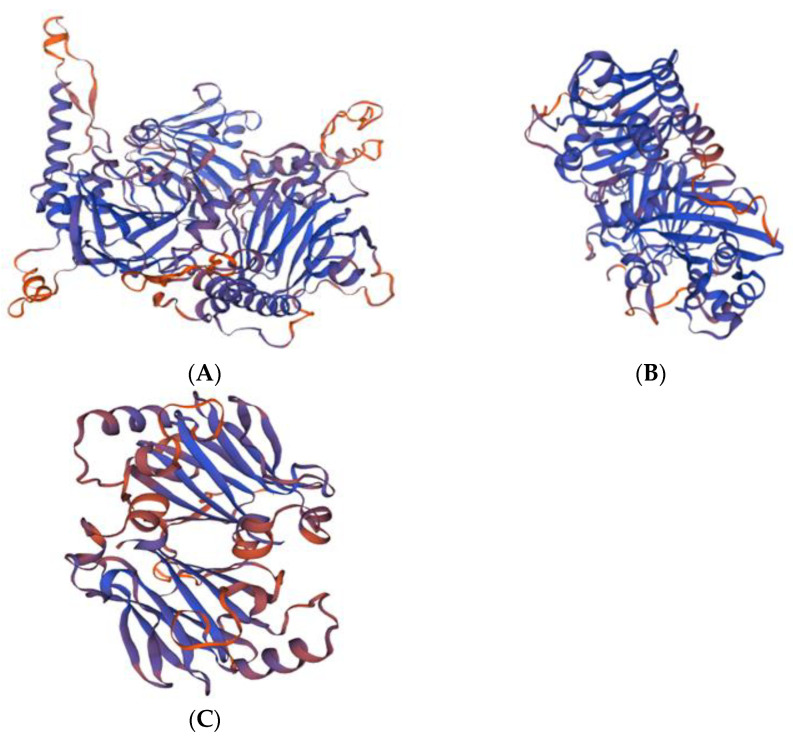
Three-dimensional tertiary structures of SMAD4 (**A**), SMAD5 (**B**) and SMAD7 (**C**) were theoretically predicted using SWISS-MODEL.

**Table 1 animals-12-02232-t001:** Primer information.

Gene	Primer Name	Primer Sequence (5′~3′)	Product Length/bp	Tem/°C
*SMAD4*	*SMAD4*-CDS1-F *SMAD4*-CDS1-R	CGAATACACCAACAAGTAATGATGC AACACCTTCGGTGTCCTTACAA	648	61
*SMAD4*	*SMAD4*-CDS2-F *SMAD4*-CDS2-R	ATCCAGCATCCACCAAGTAATCG AGTGGAAATGTAAGGTTGACGTGTG	641	64
*SMAD4*	*SMAD4*-CDS3-F *SMAD4*-CDS3-R	CTGGAGGAGATCGCTTTTGCTTG TTCCGACACCCAGCCGTTATC	582	65
*SMAD5*	*SMAD5*-CDS-S *SMAD5*-CDS-A	CCCTTCCTTTTAAATTGCGAC AATACTTACTCTGCACCGTTC	1667	55
*SMAD7*	*SMAD7*-CDS-S *SMAD7*-CDS-A	CCCGACTTCTTCATGGTGT TGTTCTGCCAACCATACCAC	1408	61
*SMAD4*	*SMAD4*-expression-S *SMAD4*-expression-A	ACGGAAGGCTTCAGGTGGCTG AGGCCACCTCCAGAGACGGG	71	64
*SMAD5*	*SMAD5*-expression-S *SMAD5*-expression-A	TCCCAGCCCATGGATACAAGCA GGGCTCTTCATAGGCGACAGGC	93	63
*SMAD7*	*SMAD7*-expression-S *SMAD7*-expression-A	ATGCTGTGCCTTCCTCCGCT CCACGCACCAGTGTGACCGA	111	64
*GAPDH*	*GAPDH*-expression-S *GAPDH*-expression-A	GCGAGATCCTGCCAACATCAAGT CCCTTCAGGTGAGCCCCAGC	105	64

**Table 2 animals-12-02232-t002:** In Tibetan sheep, the genotype and allele frequencies of single nucleotide polymorphism loci of *SMAD5* and *SMAD7*.

Gene	Locus	SNP Type	Genotype	Genotype Frequency(no.)	Allele	Allele Frequency	Exon
*SMAD5*	g.51537A>G	Synonymous type	*AA*	0.59(256)	A	0.76	3
			*AG*	0.35(150)	G	0.24	
			*GG*	0.06(27)			
*SMAD7*	g.319C>T	Synonymous type	*CC*	0.94(409)	C	0.97	3
			*CT*	0.06(24)	T	0.03	

**Table 3 animals-12-02232-t003:** Population genetic analysis of single nucleotide polymorphism loci of *SMAD5* and *SMAD7* in Tibetan sheep.

Gene	Locus	Homozygosity (*Ho*)	Heterozygosity (*He*)	Effective Allele Numbers (*Ne*)	Polymorphic Information Content (*PIC*)	χ^2^ Test
*SMAD5*	g.51537A>G	0.64	0.36	1.56	0.30	0.43
*SMAD7*	g.319C>T	0.95	0.05	1.06	0.05	1.00

*df* = 1, χ^2^_0.05_ = 3.841; *df* = 2, χ^2^_0.05_ = 5.991.

**Table 4 animals-12-02232-t004:** Association analysis of single nucleotide polymorphism of *SMAD5* and *SMAD7* and the litter size of Tibetan sheep.

Gene	Locus	Genotype	No. of Individuals	Litter Size
*SMAD5*	g.51537A>G	*AA*	256	1.07 ± 0.26
		*AG*	150	1.07 ± 0.27
		*GG*	27	1.00 ± 0.00
*SMAD7*	g.319C>T	*CC*	409	1.07 ± 0.25
		*CT*	24	1.08 ± 0.28

## Data Availability

All the data analyzed in this study are available from corresponding author.

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
