# Peer review of "Molecular Characterization, Expression Profiles of SMAD4, SMAD5 and SMAD7 Genes and Lack of Association with Litter Size in Tibetan Sheep"

_animals, 2022, doi:10.3390/ani12172232_

Round 1

Reviewer 1 Report (Previous Reviewer 4)

The revised manuscript (previously Animals-1725382)  by Sun et al. aiming at characterizing SMAD family signaling genes and their polymorphisms in the Tibetan sheep breed was largely improved for English language, and I acknowledge the authors. However, many replies to my previous comments do not fill my expectation. 

1.     The introduction part is better constructed, but the link between SMAD and litter size is still not obvious. Authors have to introduced the role of the BMP signaling pathway in the control of ovulation rate and litter size in sheep, illustrated by the numerous loss-of-function mutations described in BMP15, GDF9 and BMPR1B known as fecundity major genes. This justifies the work on SMADs as signaling effectors of BMP, knowing that SMADs may have an impact on the ovarian function as demonstrated in other species.

2.     My comment about ovary-specific Smad-interacting proteins is not solved. The reference 15 cited at the end of the sentence (lines 82-83) is not related to ovary. Remove this sentence or really cite ovary-specific proteins that functionally interact with SMADs.

3.     I can understand the limited financial context, but this is not a scientific justification to my comment about the choice of SMAD genes. If authors had fund only of the analysis of the expression and sequencing of only 3 genes, and with a good knowledge of the topic, they could have justified at least the analysis of the BMP pathway using SMAD1 (SMAD5 is also relevant), SMAD4 and SMAD6 as a specific inhibitory SMAD for the BMP signaling.

4.     Concerning the phenotypic data (LS records), I have no doubt that authors got the data and that it is real. However, authors could provide a full table (or a summary table) of individual LS record, with age and parity. As said in my previous comment, a relevant and robust evaluation of LS performance is usually based on at least 3 LS records.

5.     About qPCR, reference genes and statistical analysis, authors indicated in their answer the use of different other references genes (at least ACTB). Please indicate this in the manuscript and provide the evaluation of these genes in supplemental data to convince the reader that GAPDH was the only genuine reference gene to use in this experiment. Moreover, with the supplemental paragraph (lines 167-169), it is still unclear if the expression results were calculated with the 2-∆Ct method (direct relative expression to GAPDH) or with the 2-∆∆Ct method (relative to GAPDH and relative to a specific sample, ie. a calibrator). Looking at the figures, it seems that the 2-∆Ct method was used. Finally, the statistical analysis of expression data is still not indicated as requested in my previous comment (It should have been one-way ANOVA).

6.     Concerning the statistical model used for genotype association with LS, the levels for each parameter is still not provided. 

7.     Regarding my previous comment n°13, the fact that authors provided 3 cDNA sequences from Tibetan sheep (that is an Ovis aries breed) is not a valuable contribution to the field of genetic determinism of LS variation in sheep. It could have been interesting in the genetic diversity field if whole genome sequences were provided and analyzed. I thus still maintain my comment. The term ovis aries for Tibetan sheep is confusing. Use instead Tibetan sheep in each figure. 

8.     The new paragraph lines 471-477 are unclear. The term “correlated” should be removed and the paragraph must clearly stand that “there is no association between genotypes at SMAD5 and SMAD7 loci with LS variation in Tibetan sheep.”. This was well-done in the conclusion part. 

Author Response

Dear reviewer:

Thank you again for your nice comments on our paper. As you are concerned, there are several problems that need to be addressed. According to your nice suggestions, we have made extensive revisions to our previous manuscript.

The detailed point-by-point responses are listed in the attachment.

Reviewer 2 Report (Previous Reviewer 1)

The authors have made extensive revisions of the manuscript addressing all he issues raised by this reviewer.

Author Response

  我谨代表所有投稿作者,对您对我们文章的建设性意见表示诚挚的感谢。这些评论都是有价值的,有助于改进我们的文章。

Round 2

Reviewer 1 Report (Previous Reviewer 4)

I carefully checked the second revision of the manuscript Animals-1835401_rev2 by Sun et al. If most of the corrections made on the form are acceptable (however with introduction of errors), the comments on the scientific substance were still not taken into account. It is still unclear how many LS records were obtained per animal. The new supplementary table provided does not indicate mean LS, and let us think that some ewes have only one record that is not acceptable. Concerning qPCR analysis, authors now stand clearly, they have applied the 2-∆∆Ct method, but there is still no information on the calibrator/reference sample used for the calculation, no information on amplification efficiency. Once again, the statistical analysis for qPCR data is not described and my comment on reference gene is not solved. 

 qPCR that does not fit the gold standard (only one reference gene not controlled for validity as a reference gene in this data set, no indication of amplification efficiency and how this was measured, no information about the statistical analysis in the manuscript)

Specific comments:

1.     Gene and protein names should follow nomenclature for sheep, i.e. italicized capitalized for gene name, capitalized for protein names all along the manuscript.

2.     Line 28. This sentence about genotyping has no verb. 

3.     Lines 89-96. This new paragraph (only one sentence !!!) has to be revised for English. Was does mean “effects on lambing trait are unclear”? Prolificacy, and thus LS, was the first trait associated with these major genes and their mutations. Association with ovulation rate and ovarian function was observed secondarily.

4.     Lines 174-177. As previously explained, ∆Ct corresponds to expression relative to the reference gene(s) for each sample. ∆∆Ct corresponds to expression relative to a calibrator/reference/control sample and to the reference gene(s). Please indicate the right way you have made your analysis. Still no indication of the statistical analysis used to compare gene expression in the numerous tissues. 

Author Response

Dear reviewer,

Thanks for your professional review work on our article, point-by-point responses are in the attachment.

Round 3

Reviewer 1 Report (Previous Reviewer 4)

Once again, I carefully checked the third revision of the manuscript Animals-1835401_rev3 by Sun et al. I acknowledge the authors for the edition of the text following my last comments and for providing the supplementary table showing LS records. 

Please add the LS records information in the main text line 111, for example: Their mean litter size was calculated from 3 consecutives records from the first to the third parity (excepted for 17 ewes with only 2 records, see supplementary table 1).

For qPCR, and for the third time, I do not know how to explain better my meaning. Even if in their response to my comment, authors have copy-paste the text form Livak and Schmittgen 2001, they have forgotten to read the following and the most important paragraph talking about calibrator that I also copy-paste below. 

“The choice of calibrator for the 2∆∆CT method depends on the type of gene expression experiment that one has planned. The simplest design is to use the untreated control as the CALIBRATOR. Using the 2∆∆CT method, the data are presented as the fold change in gene expression normalized to an endogenous reference gene and relative to the untreated control. For the treated samples, evaluation of 2∆∆CT indicates the fold change in gene expression relative to the untreated control. “ 

So, if authors have used the 2∆∆Ct calculation approach as they claim, they MUST indicate the calibrator sample used (one of the tissue samples, or a mixed sample of all tissues). But, if I refer to the figures, I think authors have only used the 2∆Ct calculation giving the relative expression of the target gene compared to the reference gene (see my comment n°5 of my first revision). SO, indicate the real approach used. Moreover, amplification efficiency MUST appears for each primer pairs in the Table1. I appreciate the information added about the use of One-way ANOVA for the statistical analysis.  

Author Response

Dear reviewer:

Thanks for your comments and suggestions.

Our reply is in the attachment, please see the attachment!

This manuscript is a resubmission of an earlier submission. The following is a list of the peer review reports and author responses from that submission.

Round 1

Reviewer 1 Report

The science in the paper is decent and I think it should be published. However, the English language quality of the paper is too poor as it is. Authors should have an English speaking colleague help with the manuscript or use an editorial service to clean up the manuscript before acceptance.

Change title to the following to reflect the results of the study.

Molecular characterization, expression profiles of Smad4, Smad5 and Smad7 genes and lack of association with litter size in Tibetan sheep

Abstract

The results provided novel insights into the molecular characterization, expression profiles and polymorphisms of Smad4, Smad5 and Smad7 in Tibetan sheep but our results do not support associations of these genes and litter size of Tibetan sheep.

Introduction

Lines 50 - 53

Break up this sentence

Lines 54 - 61

Rewrite section with more clarity and better English grammar.

Materials and Methods

Please provide the approval number of the Ethics Statement. All ethics statements have approval numbers.

Line 101

You cannot start a new sentence with a number... (433 Tibetan sheep....)

Lines 146 - 149

Using the dbSNP database (http://www.ncbi.nlm.nih.gov/snp), Smad4, Smad5 and Smad7 genes SNPs were screened and verified by DNA sequencing. Improved multiplex ligation detection reaction (iMLDRTM) was used for genotyping following the instrument operating guidelines.

Statistical analysis

Lines 150 - 158

The Combination term should be clearly defined. Which variables were combined? A better term is interaction. The statistical analysis is not rigorous enough and does not account for genetic relationships between the animals (numerator relationship matrix) and is not based on true phenotypes because they were not adjusted for systematic effects (such as season, family and contemporary groups). Field observations do not represent true phenotypes in animal breeding. These should be discussed and incorporated into the discussion at the very minimum if the entire data cannot be analyzed using an animal model using restricted maximum likelihood methods.

Conclusion

Lines 512 - 513

Our study revealed that the two SNPs were not significantly associated with litter size in Tibetan sheep in this study.

Reviewer 2 Report

The manuscript is really hard to follow, it has to be corrected by a native speaker before being evaluated again.

Furthermore, the procedures are not clearly described (statistical analyses in gene expression?), so we cannot evaluate the results and the discussion. 

Reviewer 4 Report

The manuscript by Sun et al. aims at characterizing the expression profiles of SMAD family signaling genes, cloning and sequencing their cDNA and identifying their polymorphisms in the Tibetan sheep breed. As SMAD genes are part of the BMP signaling system known to control ovulation rate in sheep, authors have made an association study between the only two identified polymorphisms and litter size recorded from 433 ewes. 

As general comments, the manuscript needs large improvement of English language, and also needs improvement of the scientific level with many lacking information and overinterpretation of the data (see my specific comments below). 

Specific comments:

1.     Line 85 (also line 475). What does mean ovary-specific Smad-interacting proteins? Any gene name to cite? Please provide references.

2.     Line 90. Please provide a reference for the relation between SMADs and ovarian activity. 

3.     Lines 91-94. The goal of the paper focused on SMAD4-5 and 7 (why not SMAD2-3 and 6?) is not clear and not really supported by the introduction.

4.     Lines 98-99. Any approval number to provide?

5.     Lines 104-105. Please provide details on the LS phenotype. The number of LS record per animal? Parity? A relevant and robust evaluation of LS performance is based on at least 3 LS records from multiparous ewes.

6.     Line 110. Does it mean that others are not from purebred Tibetan sheep? 

7.     Lines 142-144. Only one reference gene is not enough for this kind of study comparing numerous tissues. If the expression of the choosen reference gene changed in some tissues, this may largely alter the analysis and the conclusion. Please use at least three references genes, test them as good reference genes by Bestkeeper and/or Normfinder algorithms for example. Please provide amplification efficiency for each primer pair and the way to obtain this efficiency. Moreover, since the 2-∆∆Ct method was applied, indicate the calibrator sample.

8.     Lines 146-147. Sentence nor clear. There is no reference to dbSNP all along the manuscript. 

9.     Line 148. Please indicate that the genotyping was done on the resting 430 animals and describe the probes used.  

10.  Line 150. There is no information on expression data statistical analysis. Please complete. 

11.  Lines 151-158. Please clarify this paragraph and provide the levels of each parameter, genotype (2 or 3 levels depending on the SNP), breeding effect (how many levels?), parity (how many levels?), sex (2 levels). Does combination =interaction? 

12.  Line 196. It should be “three” genes not “two”.

13.  Lines 216-408. To my point of view, all thus part of the results is not scientifically relevant. The main conclusion is that a known Ovis aries breed matches with Ovis aries !!! The only result is that there is no difference with the Ovis aries reference sequence excepted for one SNP in SMAD5 and one SNP in SMAD7, both leading to synonymous substitution. Alignment, phylogenetic trees, and protein structure are already available in many databases (as Ensembl), and this manuscript does bring any new relevant information. 

14.  Lines 441-444. The use of “correlated” in this sentence is confusing. Please reword.

15.  Lines 481-482. For sure!!! Phylogenetic analyses are based on sequence similarities!!! Please reword.

16.  Lines 484-489. Paragraph totally unclear.

17.  Lines 492-493. This is not demonstrated by the present work. Usually, this variant are functionally silent.   

18.  Lines 497-500. Due to poor English language skills, the concluding paragraph is confusing, letting the reader think that there is an association between the SNP evidenced and LS, while it is not the case. Please reword.